# Adolescent–Caregiver Agreement Regarding the School Bullying and Cyberbullying Involvement Experiences of Adolescents with Autism Spectrum Disorder

**DOI:** 10.3390/ijerph20043733

**Published:** 2023-02-20

**Authors:** Tai-Ling Liu, Yi-Lung Chen, Ray C. Hsiao, Hsing-Chang Ni, Sophie Hsin-Yi Liang, Chiao-Fan Lin, Hsiang-Lin Chan, Yi-Hsuan Hsieh, Liang-Jen Wang, Min-Jing Lee, Wen-Jiun Chou, Cheng-Fang Yen

**Affiliations:** 1Department of Psychiatry, School of Medicine, College of Medicine, Kaohsiung Medical University, Kaohsiung 80708, Taiwan; 2Department of Psychiatry, Kaohsiung Medical University Hospital, Kaohsiung 80708, Taiwan; 3Department of Psychology, Asia University, Taichung 41354, Taiwan; 4Department of Healthcare Administration, Asia University, Taichung 41354, Taiwan; 5Department of Psychiatry and Behavioral Sciences, University of Washington School of Medicine, Seattle, WA 98195, USA; 6Department of Psychiatry, Seattle Children’s, Seattle, WA 98105, USA; 7Department of Psychiatry, Chang Gung Memorial Hospital at Linkou, Taoyuan City 33305, Taiwan; 8School of Medicine, Chang Gung University, Taoyuan City 33302, Taiwan; 9Department of Child and Adolescent Psychiatry, Kaohsiung Medical Center, Chang Gung Memorial Hospital, Kaohsiung 83301, Taiwan; 10Department of Psychiatry, Chiayi Chang Gung Memorial Hospital, Chiayi 61363, Taiwan; 11College of Professional Studies, National Pingtung University of Science and Technology, Pingtung 91201, Taiwan

**Keywords:** autism spectrum disorder, cyberbullying, cross-informant agreement, school bullying, psychological well-being

## Abstract

School bullying and cyberbullying victimization and perpetration are prevalent in adolescents with autism spectrum disorder (AASD). However, the levels of adolescent–caregiver agreement regarding the bullying involvement of AASD and the factors associated with these levels remain to be evaluated. In the present study, we evaluated the levels of adolescent–caregiver agreement on the school bullying and cyberbullying involvement experiences of AASD and the factors associated with the levels of agreement. This study included 219 dyads of AASD and their caregivers. The school bullying and cyberbullying involvement experiences of the participating AASD were assessed using the School Bullying Experience Questionnaire and the Cyberbullying Experiences Questionnaire, respectively. Attention-deficit/hyperactivity disorder, oppositional defiant disorder (ODD), depressive and anxiety symptoms, and autistic social impairment were also assessed. AASD and their caregivers had poor to fair levels of agreement regarding the school bullying and cyberbullying victimization and perpetration experiences of AASD. Severe inattention, hyperactivity–impulsivity, ODD, depressive and anxiety symptoms, and autistic social impairment were associated with high levels of adolescent–caregiver agreement. When assessing the bullying involvement experiences of AASD, mental health professionals should obtain information from multiple sources. In addition, the factors influencing the levels of agreement should be considered.

## 1. Introduction

### 1.1. Bullying Involvement in Adolescents with Autism Spectrum Disorder

School bullying victimization and perpetration are prevalent in adolescents with autism spectrum disorder (AASD). A meta-analysis of the results of 34 studies on students with autism spectrum disorder (ASD) found that the pooled prevalence estimates for victimization, perpetration, and perpetration–victimization were 67%, 29%, and 14%, respectively; the risk of victimization in students with ASD was significantly higher than that in typically developing students and students with other disabilities [1]. Another meta-analysis found that 13% of individuals with ASD reported the experience of victimization of cyberbullying [2]. A longitudinal study conducted in AASD aged 13–15 years indicated that the experience of bullying victimization may persist during middle adolescence [3], indicating that bullying victimization can be a persistently suffering experience for AASD.

Cross-sectional studies have revealed that bullying victimization is strongly associated with internalizing (e.g., anxiety, depression, and somatic complaints) and externalizing (e.g., rule-breaking and aggressive behaviors) mental health problems [4,5,6]; furthermore, bullying perpetration is strongly associated with emotional regulation and problem externalization [6,7]. A prospective study reported that experiences of bullying victimization predicted an increase (after 12 months) in the internalization of mental health problems in children with ASD [5]. A review study found that bullying victimization can exacerbate the effects of genetic, environmental, cognitive, and physiological/neurobiological mechanisms on the development of depression in individuals with ASD [8]. In addition to traditional forms of bullying—such as physical, social, and verbal bullying—cyberbullying is also prevalent in AASD and can result in mental health problems [9]. Therefore, school bullying and cyberbullying warrants early detection and intervention to prevent subsequent mental health problems in AASD.

### 1.2. Adolescent–Caregiver Agreement Regarding Bullying Involvement

Severe autism symptoms increase the risk of bullying involvement in individuals with ASD [5]. Impaired intention attribution and theory of mind (ToM) performance are the core ASD deficits [10]. Therefore, whether AASD can accurately identify and report their bullying involvement experiences remains debatable [11,12,13,14]. For example, in the study by van Roekel et al. [13], although AASD made few mistakes in identifying bullying in video fragments, considerable discrepancies were noted between AASD-reported, teacher-, and peer-reported school bullying victimization and perpetration in real life scenarios.

Mental health professionals may require information provided by caregivers of AASD to assess the bullying experiences of AASD in clinical and counselling settings. However, the level of agreement between the information provided by AASD and that provided by their caregivers regarding their bullying victimization and perpetration experiences warrants further research. In one study, adolescents with typical development and their mothers exhibited fair to moderate levels of agreement regarding their school bullying involvement experiences; by contrast, no strong agreement regarding bullying involvement and its social effects was noted between AASD and their mothers [14]. To the best of our knowledge, no study has explored adolescent–caregiver agreement regarding the cyberbullying experiences of AASD. Cyberbullying occurs in the cyberspace; thus, the lack of nonverbal cues from facial expressions, speaking manners, and the ambience may increase the difficulty for AASD in judging the occurrence of cyberbullying. Noticing the involvement of AASD in cyberbullying is even more difficult for caregivers of these individuals because in most cases, these caregivers are unaware of the activities of AASD in the cyberspace.

### 1.3. Factors Related to Adolescent–Caregiver Agreement Regarding Bullying Involvement

Few studies have explored the factors influencing the levels of adolescent–caregiver agreement regarding the school bullying and cyberbullying involvement experiences of AASD. Older aged adolescents and girls as well as more severe ASD symptoms, social skill deficits, attention-deficit/hyperactivity disorder (ADHD), oppositional defiant disorder (ODD), depression, and anxiety are strongly associated with a higher likelihood of school bullying involvement in AASD [5,6,7,15,16] and children with typical development [17,18]. A longitudinal study revealed that social and communication skills predicted higher rates of bullying and victimization in students with ASD [19]. A study examining the associations between six dimensions of difficulties in social behaviors and bullying victimization among children with ASD found that not being optimally tuned to the social situation and resistance to changes were significantly associated with the experience of bullying victimization [20]. A meta-analysis on students with ASD concluded that deficits in social interaction and communication, externalizing symptoms, internalizing symptoms, and integrated inclusive school settings were related to higher victimization; externalizing symptoms were related to higher perpetration [1]. Moreover, a meta-analysis on 96 studies demonstrated that 28% of individuals with ASD had comorbid ADHD; 20% had anxiety disorders; 12% had disruptive, impulse-control, and conduct disorders; and 11% had depressive disorders [21]. Comorbidities, such as ADHD [22] and social anxiety [23], may exacerbate the impairments in adaptive function and social ability in children with ASD. However, whether these factors are also associated with the level of adolescent–caregiver agreement regarding the bullying involvement of adolescents remains to be investigated. Identifying such associations may facilitate understanding of the perspectives of adolescents and their caregivers regarding the bullying involvement of adolescents as well as the development of effective interventions.

### 1.4. Study Aims

The present study evaluated the levels of adolescent–caregiver agreement regarding the school bullying and cyberbullying involvement experiences of AASD and the factors associated with these levels. We hypothesized that AASD and their caregivers would have low levels of agreement. We further hypothesized that factors such as adolescent demographics, ADHD, ODD, depressive and anxiety symptoms, and autistic social impairment would be associated with the level of adolescent–caregiver agreement regarding the bullying involvement of AASD.

## 2. Methods

### 2.1. Participants and Procedures

AASD and their caregivers were enrolled from the outpatient clinics of three hospitals providing clinical services for children and adolescents with mental health problems in Taiwan between November 2010 and October 2012. The inclusion criteria for AASD included an age of 11–18 years and a diagnosis of ASD by a board-certified child psychiatrist based on the criteria outlined in the *Diagnostic and Statistical Manual of Mental Disorders*, *Fourth Edition*, *Text Revision* [24]. AASD with a comorbid intellectual disability or major psychiatric disorder (e.g., a bipolar spectrum disorder or psychiatric disorder due to physical problems) that could have prevented them from understanding the purpose and procedure of the present study were excluded. The only inclusion criterion for caregivers was being the primary caregiver of an AASD; the exclusion criteria included having cognitive impairment (e.g., addictive substance use, schizophrenia, and intellectual disability) that could have prevented them from understanding the purpose and procedure of this study. This study was approved by the Institutional Review Boards of Kaohsiung Medical University (approval number: KMUHIRB-20120084) and Chang Gung Memorial Hospital (approval number: 102e1665A3).

### 2.2. Measures

#### 2.2.1. School Bullying and Cyberbullying Involvement

The school bullying victimization and perpetration experiences of the participating AASD within the preceding month were assessed using the Chinese version of the 16-item School Bullying Experience Questionnaire (SBEQ) [25,26]. The SBEQ evaluates eight types of experiences related to social, verbal, and physical bullying victimization and perpetration in AASD. Each item is rated on a 4-point scale with endpoints ranging from 0 (*never*) to 3 (*all the time*). Higher total scores on the first and final eight items of the questionnaire indicate more severe school bullying victimization and perpetration, respectively. The cyberbullying victimization and perpetration experiences of the participating AASD within the preceding month were assessed using the 6-item Cyberbullying Experiences Questionnaire (CEQ) [27]. The CEQ evaluates three types of cyberbullying victimization and perpetration experiences of AASD. Each item is rated on a 4-point scale with endpoints ranging from 0 (*never*) to 3 (*all the time*). Higher total scores on the first and final three items of the questionnaire indicate more severe cyberbullying victimization and perpetration, respectively. The validity and reliability of the SBEQ and CEQ in Taiwan are satisfactory [26,27]. In this study, we invited AASD to report their own experiences regarding school bullying and cyberbullying involvement and invited their caregivers to rate the bullying involvement of the care recipients.

#### 2.2.2. Autistic Social Impairment

The participating caregivers rated the severity of autistic social impairment in the care recipients using the Taiwanese version of the 60-item Social Responsiveness Scale (SRS) [28,29]. Each item is rated on a 4-point scale with endpoints ranging from 1 (*never*) to 4 (*always*). Higher total scores indicate higher levels of autistic social impairment in the dimensions of social communication, manneristic behaviors, social awareness, and social emotion. The validity and reliability of the SRS, including its Taiwanese version, are satisfactory [28,29].

#### 2.2.3. ADHD and ODD

The participating caregivers rated the inattention, hyperactivity–impulsivity, and ODD symptoms observed in the care recipients within the preceding month. These symptoms were assessed using the Chinese version of the 26-item Swanson, Nolan, and Pelham Scale version IV (SNAP-IV) [30,31], where each item is rated on a 4-point scale with endpoints ranging from 0 (*not at all*) to 3 (*very much*). Higher total scores for the subscales indicate higher severity levels of caregiver-reported inattention, hyperactivity–impulsivity, and ODD. The validity and reliability of the SNAP-IV, including its Taiwanese version, are satisfactory [30,31].

#### 2.2.4. Depression

The participating AASD self-reported the depressive symptoms that they had experienced within the preceding month using the Taiwanese version of the 27-item Children’s Depression Inventory (CDI) [32,33], where each item is rated on a 3-point scale with endpoints ranging from 0 to 2. Higher total scores indicate higher severity of depressive symptoms. The validity and reliability of the CDI, including its Taiwanese version, are satisfactory [32,33].

#### 2.2.5. Anxiety

The participating AASD self-reported the anxiety symptoms that they had experienced within the preceding month using the Taiwanese version of the 39-item Multidimensional Anxiety Scale for Children (MASC) [34,35], where each item is rated on a 4-point scale with endpoints ranging from 0 (*never true about me*) to 3 (*often true about me*). Higher total scores indicate higher severity of anxiety symptoms. The validity and reliability of the MASC, including its Taiwan version, are satisfactory [34,35].

### 2.3. Statistical Analyses

Data regarding sex were expressed in number and percentage values, and data regarding age, school bullying and cyberbullying involvement, ADHD, ODD, depressive and anxiety symptoms, and autistic social impairment were expressed in mean and standard deviation values. Using intraclass correlation (ICC) we evaluated the levels of agreement between AASD- and caregiver-reported information regarding the school bullying and cyberbullying victimization and perpetration experiences of AASD. According to Cicchetti [36], ICC values ranging from 0.00 to 0.39, 0.40 to 0.59, 0.60 to 0.74, and 0.75 to 1.00 indicate that the levels of adolescent–caregiver agreement are poor, fair, good, and excellent, respectively.

We further examined the factors associated with the levels of adolescent–caregiver agreement in multiple groups of AASD. We categorized the participating AASD according to their demographics, ADHD, ODD, depressive and anxiety symptoms, and autistic social impairment. For continuous variables, the median score was used to divide the AASD into low- and high-scoring groups. Then, the differences between the groups (girl and boy groups; low- and high-score groups) in terms of the ICC were investigated. Fisher’s transformation was used to convert the sampling distributions of the ICCs and their differences into a normal distribution [37]. Next, we computed Fisher’s *z* test statistics and their corresponding *p* values. Because multiple comparisons were conducted, a *p* value of < 0.00625 (0.05/8) was regarded as significant.

## 3. Results

This study analyzed 219 dyads of AASD and their caregivers. Table 1 presents the age, sex, school bullying and cyberbullying involvement, ADHD, ODD, depressive and anxiety symptoms, and autistic social impairment statistics of the AASD cohort. Of the AASD, 27 (12.3%) were girls, and 192 (87.7%) were boys, and their mean age was 13.7 ± 2.1 years. The proportions of AASD-reported school bullying victimization and perpetration were 2.8% and 2.4%, respectively; the corresponding caregiver-reported values were 4.3% and 2.9%, respectively. The proportions of AASD-reported cyberbullying victimization and perpetration were 0.7% and 0.4%, respectively; the corresponding caregiver-reported values were 0.7% and 0.6%, respectively.

Table 2 lists the ratings corresponding to the levels of adolescent–caregiver agreement regarding the school bullying and cyberbullying involvement experiences of the AASD. The levels of agreement ranged from 0.353 to 0.435, indicating poor to fair levels of agreement.

Table 3 summarizes the differences among the AASD groups in the levels of adolescent–caregiver agreement. The levels of adolescent–caregiver agreement regarding school bullying victimization were high among the AASD with severe anxiety symptoms and autistic social impairment; by contrast, the levels of agreement regarding school bullying perpetration were high among the AASD with severe inattention, hyperactivity–impulsivity, ODD, and autistic social impairment. The levels of adolescent–caregiver agreement regarding cyberbullying victimization were also high among the AASD with severe inattention and ODD; by contrast, the levels of agreement regarding cyberbullying perpetration were high among the AASD with severe inattention, ODD, depressive symptoms, and autistic social impairment.

## 4. Discussion

In the present study, the AASD and their caregivers exhibited poor to fair levels of agreement regarding the school bullying and cyberbullying victimization and perpetration experiences of the AASD. Severe inattention, hyperactivity–impulsivity, ODD, depressive and anxiety symptoms, and autistic social impairment were all associated with high levels of adolescent–caregiver agreement regarding multiple types of bullying involvement.

### 4.1. Adolescent–Caregiver Agreement Regarding AASDs Bullying Involvements

Poor to fair levels of adolescent–caregiver agreement regarding the bullying involvement experiences of AASD indicated that the AASD and their caregivers differentially observed and interpreted interactions between AASD and peers in schools and the cyberspace. This result might be explained based on the characteristics of AASD and their caregivers. On the part of AASD, core ASD deficits, such as impaired intention attribution and ToM performance, may limit the ability of AASD to comprehend others’ behavioral intentions and thus, may increase the likelihood of misinterpretation. In one study, van Roekel et al. [13] indicated that AASD who are frequently bullied are more likely to misinterpret nonbullying situations as bullying situations; furthermore, AASD who often bully others are more likely to misinterpret bullying situations as nonbullying situations. Moreover, a study found that AASD may have difficulty in emotion recognition and hostile attribution bias, which increase their risk of verbal and covert aggression toward others [38]. However, another study observed that the ASD bullying perpetrators performed significantly better on rating the intensity of emotions in the Facial Emotion Recognition Task; the bullying victims performed significantly worse on ranking the intensity of facial emotions [39]. The discrepancy between previous studies indicates that the ability to recognize bullying involvement is heterogeneous among individuals with ASD. In addition, AASD may have difficulty clearly describing their experiences of being bullied to their caregivers and teachers, and such adolescents may refrain from reporting victimization if they believe that doing so will be in vain. AASD may also cover up their bullying perpetration to avoid being scolded. Thus, caregivers and teachers may fail to notice the experiences of bullying victimization of AASD.

On the part of caregivers, caregivers are not usually at the scene of school bullying or cyberbullying; thus, they are often not aware of bullying incidents at the time; this element is particularly true for the caregivers who rarely communicate with the teachers of their care recipients. Studies have found that there were discrepancies for externalizing and internalizing symptoms [40] and ASD symptoms of children [41] with ASD between the reports from parents and teachers. Moreover, a prospective study found a bidirectional relationship between the emotional quality of the parent–child relationships and the symptoms and emotional and behavioral problems of children with ASD [42]. Therefore, experiences of bullying involvement may be rarely discussed between AASD and their detached caregivers.

### 4.2. Factors Related to Adolescent–Caregiver Agreement

Positive associations were observed in the present study between severe inattention, hyperactivity–impulsivity, ODD, depressive and anxiety symptoms, and autistic social impairment and high levels of adolescent–caregiver agreement regarding multiple types of bullying involvement. These findings may be explained by the fact that caregivers of AASD with ADHD, ODD, emotional problems, or autistic social impairment may put greater effort into monitoring the daily activities, behaviors, and peer interactions of these adolescents than the caregivers of AASD without behavioral, emotional, or social interaction problems. A meta-analysis of 75 studies revealed that individuals with ASD showed multiple dimensions of social cognitive dysfunctions, such as ToM performance and emotion perception and processing [43]. Social cognitive dysfunctions may increase difficulties in peer interaction at school; caregivers of AASD with high autistic social impairment may be often called to school to deal with the AASD social interaction problems and have a better understanding of the AASD situation at school compared with caregivers of AASD without high autistic social impairment. Comorbid ADHD, ODD, anxiety and depressive disorders may also exacerbate AASDs executive, adaptive, and social dysfunctions [22,23]; caregivers of AASD with the aforementioned problems may have more opportunities to contact school staff than do caregivers of AASD without those problems. Therefore, on the basis of their own observations and information provided by school staff, caregivers may correctly notice the bullying involvement of AASD in a timely manner.

### 4.3. Implications

Bullying involvement can result in AASDs mental health problems. Reliable information sources help develop intervention programs for bullying involvement in AASD. Based on the results of this study, we proposed several suggestions for enhancing caregivers’ and AASDs reports on bullying involvement. First, to comprehensively assess the bullying involvement of AASD, mental health and educational professionals should consider information provided by both AASD and their caregivers. Second, because caregivers play a vital role in reducing the likelihood of bullying involvement for AASD, mental health and educational professionals should encourage caregivers to increase knowledge about school bullying and cyberbullying involvement. Third, caregivers should develop communication patterns with AASD and spend time on discussing about their school lives and peer interaction to early detect the likelihood of bullying involvement in their care recipients, even in those without severe behavioral, emotional, or social interaction problems. Fourth, because problem solving between caregivers and teachers is critical to maximizing AASD outcomes [44], enhancing the cooperation between caregivers and teachers is one of core works in the whole-school intervention program for school bullying [45]. Fifth, the results of the present study highlighted the need to identify comorbidities early in AASD. Mental health and educational professionals should observe the possibility of disagreement between caregivers’ and AASD reports on bullying involvement in AASD with comorbidity. Sixth, it is important to enhance AASDs social skills and ToM performance by involving AASD and their caregivers simultaneously into training programs. Research has found that the Program for the Education and Enrichment of Relational Skills [46] can improve AASD social skills, social knowledge, and number of hosted get-togethers with peers [47,48]. The Threat Assessment of Bullying Behavior in Youth Program can improve the self-esteem of children with ASD [49] and reduce the risk of cyberbullying victimization. The Theory of Mind Performance Training can reduce mother-reported bullying victimization in children and adolescents with high-functioning ASD [50].

### 4.4. Limitations

This study had some limitations. First, we did not collect the relevant data from the teachers and peers of the participating AASD; thus, whether the AASD and caregivers had misreported bullying involvement could not be investigated. Second, we excluded AASD with comorbid intellectual disabilities or major psychiatric disorders even though intellectual disabilities increase the risk of bullying victimization [51]. Thus, the levels of adolescent–caregiver agreement regarding the bullying involvement of AASD with intellectual disabilities or major psychiatric disorders and the factors influencing these levels should be investigated in future studies. Finally, we enrolled AASD and their caregivers from outpatient clinics. Thus, whether our findings can be generalized to AASD and their caregivers who have never visited a psychiatric clinic needs to be validated in the future.

## 5. Conclusions

We observed poor to fair levels of adolescent–caregiver agreement regarding the school bullying and cyberbullying involvement experiences of AASD. Mental health professionals should obtain information from multiple sources when assessing the bullying involvement experiences of AASD. ADHD, ODD, depressive and anxiety symptoms, and autistic social impairment were found to be associated with the levels of adolescent–caregiver agreement regarding bullying involvement. Mental health professionals should consider these factors when assessing the bullying involvement of AASD and developing the relevant interventions to help AASD and their caregivers manage bullying involvement.

## Figures and Tables

**Table 1 ijerph-20-03733-t001:** Age, sex, bullying involvement, ADHD, ODD, depression and anxiety symptoms, and autistic social impairment of adolescents with autism spectrum disorder (*N* = 219).

	*n* (%)/Mean ± SD
Child’s sex	
Boys	192 (87.7%)
Girls	27 (12.3%)
Age (years)	13.7 (2.1)
Victimization of school bullying	
Adolescent-reported	2.7 (2.8)
Caregiver-reported	5.8 (4.3)
Perpetration of school bullying	
Adolescent-reported	1.8 (2.4)
Caregiver-reported	3.0 (2.9)
Victimization of cyberbullying	
Adolescent-reported	0.2 (0.7)
Caregiver-reported	0.3 (0.7)
Perpetration of cyberbullying	
Adolescent-reported	0.1 (0.4)
Caregiver-reported	0.2 (0.6)
SNAP-IV	
Inattention	14.7 (6.5)
Hyperactivity–impulsivity	10.0 (6.7)
ODD symptoms	10.7 (6.3)
Depression on the CDI-TW	14.6 (10.1)
Anxiety symptoms on the MASC-T	35.2 (16.8)
Autistic social impairment on the SRS	154.5 (27.2)

ADHD: attention-deficit/hyperactivity disorder; CDI-TW: Children’s Depression Inventory, Taiwanese version; MASC-T: Multidimensional Anxiety Scale for Children, Taiwanese version; ODD: oppositional defiant disorder; SNAP-IV: short version of the Swanson, Nolan, and Pelham Scale, version IV; and SRS: Social Responsiveness Scale.

**Table 2 ijerph-20-03733-t002:** Levels of adolescent–caregiver agreement regarding the bullying involvement experiences of adolescents with autism spectrum disorder (*N* = 219).

Variables	Intra-Class Correlation
Victimization of school bullying	0.435 ***
Perpetration of school bullying	0.353 ***
Victimization of cyberbullying	0.391 ***
Perpetration of cyberbullying	0.404 ***

*** *p* < 0.001.

**Table 3 ijerph-20-03733-t003:** Differences among various groups of adolescents with autism spectrum disorder regarding levels of adolescent–caregiver agreement.

Variables	Groups			Intra-Class Correlation				
	G1-G2	G1 (*n*)	G2 (*n*)	ICC in G1	ICC in G2	Difference	z	*p*
Victimization of school bullying								
Child’s sex	Girl–Boy	27	192	0.546	0.413	0.133	0.801	0.423
Child’s age	Low–High	111	108	0.396	0.445	−0.049	−0.433	0.665
Inattention	Low–High	115	104	0.456	0.431	0.025	0.225	0.822
Hyperactivity–impulsivity	Low–High	119	100	0.325	0.486	−0.160	−1.403	0.161
ODD symptoms	Low–High	111	108	0.402	0.478	−0.075	−0.680	0.497
Depression	Low–High	112	107	0.513	0.293	0.220	1.933	0.053
Anxiety	Low–High	113	106	0.115	0.496	−0.381	−3.121	0.002
Autistic social impairment	Low–High	110	109	0.135	0.546	−0.411	−3.477	0.001
Perpetration of school bullying								
Child’s sex	Girl–Boy	27	192	0.468	0.329	0.138	0.762	0.446
Child’s age	Low–High	111	108	0.249	0.422	−0.173	−1.432	0.152
Inattention	Low–High	115	104	0.125	0.473	−0.349	−2.835	0.005
Hyperactivity–impulsivity	Low–High	119	100	0.084	0.441	−0.357	−2.831	0.005
ODD symptoms	Low–High	111	108	0.057	0.463	−0.406	−3.243	0.001
Depression	Low–High	112	107	0.146	0.395	−0.250	−1.981	0.048
Anxiety	Low–High	113	106	0.123	0.365	−0.242	−1.887	0.059
Autistic social impairment	Low–High	110	109	−0.057	0.519	−0.576	−4.611	<0.001
Victimization of cyberbullying								
Child’s sex	Girl–Boy	27	192	0.362	0.397	−0.035	−0.191	0.849
Child’s age	Low–High	111	108	0.302	0.461	−0.159	−1.365	0.172
Inattention	Low–High	115	104	0.030	0.705	−0.674	−6.167	<0.001
Hyperactivity–impulsivity	Low–High	119	100	0.215	0.482	−0.267	−2.231	0.026
ODD symptoms	Low–High	111	108	−0.084	0.556	−0.640	−5.187	<0.001
Depression	Low–High	112	107	0.274	0.385	−0.111	−0.908	0.364
Anxiety	Low–High	113	106	0.361	0.394	−0.033	−0.281	0.779
Autistic social impairment	Low–High	110	109	0.214	0.480	−0.266	−2.227	0.026
Perpetration of cyberbullying								
Child’s sex	Girl–Boy	27	192	−0.059	0.435	−0.494	−2.421	0.015
Child’s age	Low–High	111	108	0.359	0.385	−0.026	−0.218	0.828
Inattention	Low–High	115	104	−0.106	0.488	−0.594	−4.665	<0.001
Hyperactivity–impulsivity	Low–High	119	100	0.471	0.365	0.106	0.939	0.348
ODD symptoms	Low–High	111	108	−0.078	0.466	−0.544	−4.258	<0.001
Depression	Low–High	112	107	−0.106	0.426	−0.532	−4.096	<0.001
Anxiety	Low–High	113	106	0.472	0.332	0.140	1.218	0.223
Autistic social impairment	Low–High	110	109	−0.110	0.494	−0.604	−4.755	<0.001

G1, group 1; G2, group 2; and ICC, intraclass correlation. Median values: age, 13.2 years; inattention, 15; hyperactivity/impulsivity, 9; oppositional defiant disorder (ODD), 10; depression, 12; anxiety, 34; and autistic social impairment, 155.

## Data Availability

The data used in this study are available upon reasonable request to the corresponding authors.

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
