# Peer review of "Adolescent–Caregiver Agreement Regarding the School Bullying and Cyberbullying Involvement Experiences of Adolescents with Autism Spectrum Disorder"

_ijerph, 2023, doi:10.3390/ijerph20043733_

Round 1

Reviewer 1 Report

The focus of the paper (bullying & cyberbullying) is sound and reflects a critical concern in today's world - not just for youth but for youth with ASD.  Suggestions: The Discussion section could be enhanced with more discussion regarding each of the findings and links to literature. The section also needs more discussion regarding the suggested actions to be taken (e.g., who should educate caregivers - beyond the Mental Health professionals - and also teachers regarding the effects of all types of bullying ... and what might be some suggested proven ways to assist).       

Author Response

We appreciated your valuable comments. As discussed below, we have revised our manuscript with underlines based on your suggestions. Please let us know if we need to provide anything else regarding this revision.

Comment 1

The Discussion section could be enhanced with more discussion regarding each of the findings and links to literature. The section also needs more discussion regarding the suggested actions to be taken (e.g., who should educate caregivers - beyond the Mental Health professionals - and also teachers regarding the effects of all types of bullying ... and what might be some suggested proven ways to assist). 

Response

Thank you for your comment. We rewrote the Discussion section regarding each of the findings and links to literature. We also added more discussion regarding the suggested actions to be taken. Please refer to line 260-341.

“4.1. Adolescent–Caregiver Agreement Regarding AASD’s Bullying Involvements

Poor to fair levels of adolescent–caregiver agreement regarding the bullying involvement experiences of AASD indicated that the AASD and their caregivers differentially observed and interpreted interactions between AASD and peers in schools and cyberspace. This result might be explained based on the characteristics of AASD and their caregivers. On the part of AASD, core ASD deficits, such as impaired intention attribution and ToM performance, may limit the ability of AASD to comprehend others’ behavioral intentions and thus may increase the likelihood of misinterpretation. In one study, van Roekel et al. [13] indicated that AASD who are frequently bullied are more likely to misinterpret nonbullying situations as bullying situations; furthermore, AASD who often bully others are more likely to misinterpret bullying situations as nonbullying situations. Moreover, a study found that AASD may have difficulty in emotion recognition and hostile attribution bias that increase their risk of verbal and covert aggression toward others [38]. However, another study observed that ASD bullying perpetrators performed significantly better on rating the intensity of emotion in the Facial Emotion Recognition Task; bullying victims performed significantly worse on ranking the intensity of facial emotion [39]. The discrepancy between previous studies indicates that the ability to recognizing bullying involvement is heterogeneous among individuals with ASD. In addition, AASD may have difficulty clearly describing their experiences of being bullied to their caregivers and teachers, and such adolescents may refrain from reporting victimization if they believe that doing so will be in vain. AASD may also cover up their bullying perpetration to avoid being scolded. Thus, caregivers and teachers may fail to notice the experiences of bullying victimization of AASD.

On the part of caregivers, caregivers are not usually at the scene of school bullying or cyberbullying; thus, they are often not aware of bullying incidents at the time; this element is particularly true for the caregivers who rarely communicate with the teachers of their care recipients. Studies have found that there were discrepancies for externalizing and internalizing symptoms [40] and ASD symptoms of children [41] with ASD between the reports from parents and teachers. Moreover, a prospective study found a bidirectional relationship between the emotional quality of the parent-child relationships and the symptoms and emotional and behavioral problems of children with ASD [42]. Therefore, experiences of bullying involvement may be rarely discussed between AASD and their detached caregivers.

4.2. Factors Related to Adolescent–Caregiver Agreement

Positive associations were observed in the present study between severe inattention, hyperactivity–impulsivity, ODD, depressive and anxiety symptoms, and autistic social impairment and high levels of adolescent–caregiver agreement regarding multiple types of bullying involvement. These findings may be explained by the fact that caregivers of AASD with ADHD, ODD, emotional problems, or autistic social impairment may put greater effort into monitoring the daily activities, behaviors, and peer interactions of these adolescents than do caregivers of AASD without behavioral, emotional, or social interaction problems. A meta-analysis on 75 studies revealed that individuals with ASD showed multiple dimensions of social cognitive dysfunctions such as ToM performance and emotion perception and processing [43]. Social cognitive dysfunctions may increase difficulties in peer interaction at school; caregivers of AASD with high autistic social impairment may be often called to school to deal with AASD’s social interaction problems and have a better understanding of AASD’s situation at school compared with caregivers of AASD without high autistic social impairment. Comorbid ADHD, ODD, anxiety and depressive disorders may also exacerbate AASD’s executive, adaptive and social dysfunctions [22,23]; caregivers of AASD with the aforementioned problems may have more opportunities to contact school staff than do caregivers of AASD without those problems. Therefore, on the basis of their own observations and information provided by school staff, caregivers may correctly notice the bullying involvement of AASD in a timely manner.

4.3. Implications

Bullying involvement can result in AASD’ mental health problems. Reliable information sources help develop intervention programs for bullying involvement in AASD. Based on the results of this study, we proposed several suggestions for enhancing caregivers’ and AASD’s reports on bullying involvement. First, to comprehensively assess the bullying involvement of AASD, mental health and educational professionals should consider information provided by both AASD and their caregivers. Second, because caregivers play a vital role in reducing the likelihood of bullying involvement for AASD, mental health and educational professionals should encourage caregivers to increase knowledge about school bullying and cyberbullying involvement. Third, caregivers should develop communication patterns with AASD and spend time on discussing about their school lives and peer interaction to early detect the likelihood of bullying involvement in their care recipients, even in those without severe behavioral, emotional, or social interaction problems. Fourth, because problem solving between caregivers and teachers is critical to maximizing AASD’ outcomes [44], enhancing the cooperation between caregivers and teachers is one of core works in the whole-school intervention program for school bullying [45]. Fifth, the results of the present study highlighted the need to identify comorbidities early in AASD. Mental health and educational professionals should observe the possibility of disagreement between caregivers’ and AASD’s reports on bullying involvement in AASD with comorbidity. Sixth, it is important to enhance AASD’s social skills and ToM performance by involving AASD and their caregivers simultaneously into training programs. Research has found that the Program for the Education and Enrichment of Relational Skills [46] can improve AASD’s social skills, social knowledge, and number of hosted get-togethers with peers [47,48]. The Threat Assessment of Bullying Behavior in Youth Program can improve self-esteem of children with ASD [49] and reduce the risk of cyberbullying victimization. The Theory of Mind Performance Training can reduce mother-reported bullying victimization in children and adolescents with high-functioning ASD [50].

Reviewer 2 Report

 The topic addressed by the article "Adolescent–Caregiver Agreement Regarding the School Bullying and Cyberbullying Involvement Experiences of Adolescents With Autism Spectrum Disorder is of great interest. However, the references cited and the collect data is outdated. Since 2012 when the authors finished collecting the information until the present time, more than 10 years have passed, and awareness about bullying and cyberbullying has increased.This has been able to lead to greater awareness among caregivers, and therefore in a better identification of bullying situations. In addition, the fact of not having taken into account a control group in the research makes it difficult to know if the results found would be the same in a sample without AASD.

Author Response

We appreciated your valuable comments. As discussed below, we have revised our manuscript with underlines based on your suggestions. Please let us know if we need to provide anything else regarding this revision.

Comment 1

The topic addressed by the article "Adolescent–Caregiver Agreement Regarding the School Bullying and Cyberbullying Involvement Experiences of Adolescents With Autism Spectrum Disorder is of great interest. However, the references cited and the collect data is outdated. Since 2012 when the authors finished collecting the information until the present time, more than 10 years have passed, and awareness about bullying and cyberbullying has increased. This has been able to lead to greater awareness among caregivers, and therefore in a better identification of bullying situations. In addition, the fact of not having taken into account a control group in the research makes it difficult to know if the results found would be the same in a sample without AASD.

Response

Thank you for your comments.

  1. Although this study analyzed the data collected in 2012, to the best of our knowledge, only one study has examined the levels of adolescent-caregiver agreement regarding school bullying involvement experiences of adolescent with ASD [14], and no study has explored adolescent–caregiver agreement regarding the cyberbullying experiences of adolescent with ASD. Moreover, factors that relate to the level of adolescent–caregiver agreement regarding the bullying involvement of adolescents with ASD have not been investigated. Therefore, we believe the present study had its originality and can add new knowledge to the field of intervention for bullying involvement in adolescents with ASD.
  2. A previous study has demonstrated that the levels of adolescent–caregiver agreement regarding bullying involvement were different between adolescents with ASD and their mothers and adolescents without ASD and their mothers [14]. Severe autism symptoms increase the risk of bullying involvement in individuals with ASD [5]. The risk of bullying victimization in students with ASD has been also found to be significantly higher than that in typically developing students and students with other disabilities [1]. Therefore, this study focused on adolescents with ASD and their caregivers.

Reviewer 3 Report

It's a very interesting study. I recommend to add a space in row 205 because Table 1 footnote continues with the text presentation of Table 2. Regarding Table 2, the footnote says Table 3, sort needs to be reviewed and corrected.

Other than that, the authors should include more recent references on the introduction and discussion.  I noticed that only five out of 30 references are less than 5 years old. 

Author Response

We appreciated your valuable comments. As discussed below, we have revised our manuscript with underlines based on your suggestions. Please let us know if we need to provide anything else regarding this revision.

Comment 1

It's a very interesting study. I recommend to add a space in row 205 because Table 1 footnote continues with the text presentation of Table 2. Regarding Table 2, the footnote says Table 3, sort needs to be reviewed and corrected.

Response

We apologized for the typographical errors. In the revised manuscript, we added a space between the end of footnote of Table 1 and the next paragraph describing the results of Table 2. Please refer to line 230. We also corrected the legends of Table 2 and Table 3. Please refer to line 234 and 247.

Comment 2

Other than that, the authors should include more recent references on the introduction and discussion.  I noticed that only five out of 30 references are less than 5 years old. 

Response

Thank you for your comment. We added 16 new references published in recent 5 years listed below into the revised manuscript. The contents of Introduction and Discussion were rewritten accordingly.

  1. Park, I.; Gong, J.; Lyons, G.L.; Hirota, T.; Takahashi, M.; Kim, B.; Lee, S.Y.; Kim, Y.S.; Lee, J.; Leventhal, B.L. Prevalence of and factors associated with school bullying in students with autism spectrum disorder: A cross-cultural meta-analysis. Yonsei Med. J. 2020, 61, 909-922. doi:10.3349/ymj.2020.61.11.909.
  2. Trundle, G.; Jones, K.A.; Ropar, D.; Egan, V. Prevalence of victimisation in autistic individuals: A systematic review and meta-analysis. Trauma Violence Abuse. 2022, 15248380221093689. doi:10.1177/15248380221093689.
  3. Oakley, B.; Loth, E.; Murphy, D.G. Autism and mood disorders. Int Rev Psychiatry. 2021, 33, 280-299. doi:10.1080/09540261.2021.1872506. 
  4. Libster, N.; Knox, A.; Engin, S.; Geschwind, D.; Parish-Morris, J.; Kasari, C. Personal victimization experiences of autistic and non-autistic children. Mol Autism. 2022, 13, 51. doi:10.1186/s13229-022-00531-4.
  5. Matthias, C.; LaVelle, J.M.; Johnson, D.R.; Wu, Y.C.; Thurlow, M.L. Exploring predictors of bullying and victimization of students with autism spectrum disorder (ASD): Findings from NLTS 2012. J Autism Dev Disord. 2021, 51, 4632-4643. doi:10.1007/s10803-021-04907-y.
  6. Forrest, D.L.; Kroeger, R.A.; Stroope, S. Autism spectrum disorder symptoms and bullying victimization among children with autism in the United States. J Autism Dev Disord. 2020, 50, 560-571. doi:10.1007/s10803-019-04282-9.
  7. Lai, M.C.; Kassee, C.; Besney, R.; Bonato, S.; Hull, L.; Mandy, W.; Szatmari, P.; Ameis, S.H. Prevalence of co-occurring mental health diagnoses in the autism population: a systematic review and meta-analysis. Lancet Psychiatry. 2019, 6, 819-829. doi:10.1016/S2215-0366(19)30289-5.
  8. Liu, Y.; Wang, L.; Xie, S.; Pan, S.; Zhao, J.; Zou, M.; Sun, C. Attention deficit/hyperactivity disorder symptoms impair adaptive and social function in children with autism spectrum disorder. Front Psychiatry. 2021, 12, 654485. doi:10.3389/fpsyt.2021.654485.
  9. Briot, K.; Jean, F.; Jouni, A.; Geoffray, M.M.; Ly-Le Moal, M.; Umbricht, D.; Chatham, C.; Murtagh, L.; Delorme, R.; Bouvard, M.; et al. Social anxiety in children and adolescents with autism spectrum disorders contribute to impairments in social communication and social motivation. Front Psychiatry. 2020, 11, 710. doi:10.3389/fpsyt.2020.00710.
  10. Kirst, S.; Bögl, K.; Gross, V.L.; Diehm, R.; Poustka, L.; Dziobek, I. Subtypes of aggressive behavior in children with autism in the context of emotion recognition, hostile attribution bias, and dysfunctional emotion regulation. J Autism Dev Disord. 2022, 52, 5367-5382. doi:10.1007/s10803-021-05387-w.
  11. Liu, T.L.; Wang, P.W.; Yang, Y.C.; Shyi, G.C.; Yen, C.F. Association between Facial Emotion Recognition and Bullying Involvement among Adolescents with High-Functioning Autism Spectrum Disorder. Int J Environ Res Public Health. 2019, 16, 5125. doi:10.3390/ijerph16245125. 
  12. Hickey, E.J.; Bolt, D.; Rodriguez, G.; Hartley, S.L. Bidirectional relations between parent warmth and criticism and the symptoms and behavior problems of children with autism. J Abnorm Child Psychol. 2020, 48, 865-879. doi:10.1007/s10802-020-00628-5.
  13. Velikonja, T.; Fett, A.K.; Velthorst, E. Patterns of nonsocial and social cognitive functioning in adults with autism spectrum disorder: A systematic review and meta-analysis. JAMA Psychiatry. 2019, 76, 135-151. doi:10.1001/jamapsychiatry.2018.3645.
  14. Azad, G.F.; Gormley, S.; Marcus, S.; Mandell, D.S. Parent-teacher problem solving about concerns in children with autism spectrum disorder: The role of income and race. Psychol Sch. 2019, 56, 276-290. doi:10.1002/pits.22205.
  15. Touloupis, T.; Athanasiades, C. Cyberbullying and empathy among elementary school students: Do special educational needs make a difference? Scand J Psychol. 2022, 63, 609-623. doi:10.1111/sjop.12838.
  16. Liu, M.J.; Ma, L.Y.; Chou, W.J.; Chen, Y.M.; Liu, T.L.; Hsiao, R.C.; Hu, H.F.; Yen, C.F. Effects of theory of mind performance training on reducing bullying involvement in children and adolescents with high-functioning autism spectrum disorder. PLoS One. 2018, 13, e0191271. doi:10.1371/journal.pone.0191271.

Round 2

Reviewer 2 Report

The text has now been improved.